# Obesity Is Associated with Fatty Liver and Fat Changes in the Kidneys in Humans as Assessed by MRI

**DOI:** 10.3390/nu16091387

**Published:** 2024-05-03

**Authors:** Hadar Raphael, Eyal Klang, Eli Konen, Yael Inbar, Avshalom Leibowitz, Yael Frenkel-Nir, Sara Apter, Ehud Grossman

**Affiliations:** 1Arrow Projects for Medical Research Education, The Chaim Sheba Medical Center, Tel Hashomer 5266202, Israel; aharonith@gmail.com; 2Department of Imaging, The Chaim Sheba Medical Center, Tel Hashomer 5266202, Israel; eyalkla@hotmail.com (E.K.); eli.konen@sheba.health.gov.il (E.K.); yael.inbar@sheba.health.gov.il (Y.I.); saraapter@gmail.com (S.A.); 3Faculty of Medicine, Tel Aviv University, Tel Aviv 5266202, Israel; avshalom.leibowitz@sheba.health.gov.il (A.L.); yael.frenkelnir@sheba.health.gov.il (Y.F.-N.); 4Internal Medicine D the Chaim Sheba Medical Center, Tel Hashomer 5266202, Israel; 5Medical Management Department, The Chaim Sheba Medical Center, Tel Hashomer 5266202, Israel; 6Adelson Medical School, Ariel University, Ariel 4077625, Israel

**Keywords:** obesity, fatty liver, fatty kidneys, MRI

## Abstract

Background: Obesity is associated with metabolic syndrome and fat accumulation in various organs such as the liver and the kidneys. Our goal was to assess, using magnetic resonance imaging (MRI) Dual-Echo phase sequencing, the association between liver and kidney fat deposition and their relation to obesity. Methods: We analyzed MRI scans of individuals who were referred to the Chaim Sheba Medical Center between December 2017 and May 2020 to perform a study for any indication. For each individual, we retrieved from the computerized charts data on sex, and age, weight, height, body mass index (BMI), systolic and diastolic blood pressure (BP), and comorbidities (diabetes mellitus, hypertension, dyslipidemia). Results: We screened MRI studies of 399 subjects with a median age of 51 years, 52.4% of whom were women, and a median BMI 24.6 kg/m^2^. We diagnosed 18% of the participants with fatty liver and 18.6% with fat accumulation in the kidneys (fatty kidneys). Out of the 67 patients with fatty livers, 23 (34.3%) also had fatty kidneys, whereas among the 315 patients without fatty livers, only 48 patients (15.2%) had fatty kidneys (*p* < 0.01). In comparison to the patients who did not have a fatty liver or fatty kidneys (*n* = 267), those who had both (*n* = 23) were more obese, had higher systolic BP, and were more likely to have diabetes mellitus. In comparison to the patients without a fatty liver, those with fatty livers had an adjusted odds ratio of 2.91 (97.5% CI; 1.61–5.25) to have fatty kidneys. In total, 19.6% of the individuals were obese (BMI ≥ 30), and 26.1% had overweight (25 < BMI < 30). The obese and overweight individuals were older and more likely to have diabetes mellitus and hypertension and had higher rates of fatty livers and fatty kidneys. Fat deposition in both the liver and the kidneys was observed in 15.9% of the obese patients, in 8.3% of the overweight patients, and in none of those with normal weight. Obesity was the only risk factor for fatty kidneys and fatty livers, with an adjusted OR of 6.3 (97.5% CI 2.1–18.6). Conclusions: Obesity is a major risk factor for developing a fatty liver and fatty kidneys. Individuals with a fatty liver are more likely to have fatty kidneys. MRI is an accurate modality for diagnosing fatty kidneys. Reviewing MRI scans of any indication should include assessment of fat fractions in the kidneys in addition to that of the liver.

## 1. Introduction

Obesity has grown to epidemic proportions, with over 4 million people dying each year because of being overweight or obese, according to the global burden of disease [1]. By 2030, nearly one in two adults will have obesity, and it is projected that nearly one in four adults will have severe obesity [2].

Obesity can lead to type 2 diabetes (T2D), hypertension, and dyslipidemia and is associated with metabolic syndrome (MS), which is a combination of abdominal obesity, elevated blood pressure, high triglyceride and low HDL cholesterol levels, and high fasting glucose levels [3,4]. Using a rat model, we showed that a high-fructose diet (HFD) could cause MS and fatty liver disease [5,6]. MS is a known risk factor for developing non-alcoholic fatty liver disease (NAFLD) [7]. Studies have shown that individuals with MS are at a higher risk of developing NAFLD than those without MS [8]. The exact mechanism by which MS leads to NAFLD is not fully understood; it is believed, however, that insulin resistance plays a key role. Insulin resistance can lead to increased production of free fatty acids and triglycerides, which can accumulate in the liver and cause fatty liver disease. In a recent study, we showed that a HFD could also cause fat accumulation in the kidneys [9]. Obesity can also increase the risk of developing fatty kidneys, as it is associated with insulin resistance, inflammation, and oxidative stress, which can all contribute to the development of fatty kidneys [10]. Additionally, high blood pressure, high triglyceride levels, and high blood glucose levels, all of which are components of MS, can contribute to the development of fatty kidney disease [4]. This can lead to inflammation and scarring of the kidney tissue, which can result in reduced kidney function and, in severe cases, renal failure. Fatty liver and fatty kidney disease are two distinct conditions, although they share a common underlying mechanism: the accumulation of excess fat in the respective organs. Magnetic Resonance Imaging (MRI) is an accurate method for assessing and monitoring lipid content in the liver [11,12]. This technique can be utilized to assess kidney parenchyma lipid content. The aim of the present study was two-fold. First, we sought to use a new MRI methodology (dual-echo phase sequencing) to assess the rates of fatty kidney and fatty liver disease in a large group of subjects and evaluate the association between these distinct conditions. The second aim was to evaluate the association between obesity and fatty liver and fatty kidney disease.

## 2. Methods

### 2.1. Study Population

We reviewed 695 MRI abdominal scans that were performed for any indication at the Sheba Medical Center between December 2017 and May 2020. In most cases, the indications for MRI scan were an evaluation of suspected malignancy or follow-up after the removal of an abdominal or gynecological tumor. We excluded MRI scans that were performed on pregnant women (81) or children younger than 16 (110), were of bad quality (18), and/or had no dual-echo phase scans (87). We therefore included in this study 399 MRI scans. (Figure 1). For each subject, we retrieved from computerized charts data on sex, age, weight, height, body mass index (BMI), systolic and diastolic blood pressure (BP), comorbidities (diabetes mellitus, hypertension, dyslipidemia), and renal function tests. Dyslipidemia was defined when an individual either used lipid-lowering agents or had the diagnostic ICD-9 code of dyslipidemia in his chart. We were blind with respect to each patient’s metabolic status and whether they had a known affliction with NAFLD for each scan.

### 2.2. MRI Assessment

We assessed the fat fractions of the liver and kidneys using MRI dual-echo phase sequencing, a method that is based on the relative chemical shift between water and the methylene peak of fat. Volumetric liver fat can be detected by determining the relative signal loss between in-phase (IP) and opposed-phase (OP) images. This method is one of the imaging methods accepted for identifying and characterizing liver fat fractions [12]. We employed the fat signal fraction formula, i.e., $\eta =\frac{IP-OP}{2IP}$, to evaluate fat accumulation in the kidneys [13]. We measured each kidney at three points (defined on the coronal images): the upper, middle, and lower poles. We then chose the maximal value to calculate the kidney fat fraction. As described by Krievina G. et al. [14], we found higher rates of fat accumulation in the left kidney, and therefore we focused on the left kidney for each case. An illustration of the evaluation of fat in the liver and the kidneys is presented in Figure 2.

Fatty liver was defined as liver steatosis grade 1 and above that correlated with 5% or more of the fat fraction. This definition is based on histologic studies on liver steatosis [12]. Since a fatty kidney is not clearly defined in the literature and since there is no research on the correlation between MRI findings and histologic structure in this regard, we decided to set a threshold of a fat fraction of 4% to define a fatty kidney.

### 2.3. Statistical Analysis

The significance level was set at *p* < 0.05 for all analyses. Continuous variables were expressed as medians and interquartile ranges (IQRs), while categorical variables were presented as percentages.

We compared the rates of cases of fatty livers to the rates of cases with fat changes in the kidney. Using the chi-squared test, we analyzed the association between the presence of fatty liver and fatty kidneys. We also divided the patients into 3 groups according to BMI: normal weight (BMI ≤ 25 kg/m^2^), overweight 25 < BMI < 30 kg/m^2^), and obese (BMI ≥ 30 kg/m^2^). We calculated the rates of fatty liver and kidneys in the different groups.

We then assessed the associations between these clinical variables and the presence of fatty liver, fatty kidneys, and both. Continuous variables were compared using the non-parametric Mann–Whitney U test for paired comparisons, while categorical variables were analyzed using the chi-squared test.

To further investigate the associations, we performed adjusted odds ratio (aOR) analyses using logistic regression models. First, we calculated the aOR for fatty kidneys as the target, using age, sex, diabetes, hypertension, dyslipidemia, and fatty liver as independent variables. Next, we calculated the aOR for subjects with both fatty kidney and fatty liver disease, using age, sex, diabetes, dyslipidemia, and BMI ≥ 30 as independent variables.

## 3. Results

### 3.1. Subjects’ Characteristics

We screened the MRIs of 399 subjects whose mean age was 51 ± 19 (median 51) years, consisting of 52.4% women and exhibiting a mean BMI of 25.8 ± 5.6 (median 24.6) kg/m^2^. Hypertension was identified in 86 subjects (21.6%). Diabetes mellitus and dyslipidemia were identified in less than 10% of the subjects (Table 1).

### 3.2. The rates of Fatty Liver and Fatty Kidney Disease

Among the subjects we screened, 72 (18%) had fatty livers. The subjects with fatty liver disease were more obese and were more likely to have hypertension, diabetes mellitus, and dyslipidemia than those without a fatty liver (Table 1).

For the evaluation of fatty kidneys, we excluded 17 patients who had undergone a left nephrectomy. Out of 382 patients, 71 (18.6%) had fatty kidneys. Those with fatty kidneys were more obese and had higher systolic BP levels than those without fatty kidneys (Table 2). They also had higher creatinine and urea levels (0.89 vs. 0.76 and 35 vs. 30 for creatinine and urea, respectively, with *p* < 0.05 for both).

To analyze the association between fatty liver and fatty kidneys, we included 382 patients who had undergone an evaluation of both their liver and kidneys. Figure 3 depicts a case with a fatty liver and a fatty kidney. Out of the 67 patients with fatty livers, 23 (34.3%) also had fatty kidneys, whereas among the 315 patients without a fatty liver, only 48 patients (15.2%) had fatty kidneys. In comparison to patients who had neither a fatty liver nor fatty kidneys (*n* = 267), those who had both (*n* = 23) were more obese and had higher systolic BP and were more likely to have diabetes mellitus (Table 3).

In comparison to the patients without a fatty liver, those with a fatty liver had an adjusted odds ratio of 2.91 (97.5% CI; 1.61–5.25) with respect to having fatty kidneys (Table 4).

### 3.3. The Association between Body Mass Index and Fatty Liver and Fatty Kidneys

Height and weight data were available for 322 subjects. Most subjects had normal weights, and 19.6% were obese. In the normal-weight group, 18 subjects had a very low BMI (<18.5 kg/m^2^). Inclusion or exclusion of the subgroup with a very low BMI did not affect the findings for the normal-weight group.

The overweight and obese patients were older and more likely to have diabetes mellitus and hypertension and had higher rates of fatty liver and fatty kidney disease (Table 5). Fatty deposition in both the liver and the kidneys was observed in 15.9% of the obese patients, 8.3% of the overweight patients, and none of those with normal weight (Table 5). Obesity was the only risk factor for fatty kidney and fatty liver disease, with an adjusted OR of 6.3 (97.5% CI 2.1–18.6) (Table 6).

## 4. Discussion

In this study, we used MRI to assess, in a large cohort, the fat content of the liver and kidneys and relate it to BMI.

We found a fatty liver in 18% of the individuals. This rate is slightly less than the current estimation of global prevalence, i.e., 25%, described in a meta-analysis by Younossi. Z et al. [15].

However, we used strict MRI criteria that may underestimate mild cases of fatty liver disease. MRI-proton density fat fraction is considered a reliable, non-invasive method for assessing liver steatosis and is commonly used in clinical practice [12]. The advantages of this method include its results close correlation with histopathological examination results and the convenience of the quantification of steatosis [12].

We found fatty kidneys in 18.6% of the individuals. Among 2923 participants of the Framingham Heart Study, Foster M.C. et al. [16] found fat deposits in the kidney in 30.1% of the participants, but they measured the fat deposition in the renal sinus by using CT scans and used sex-specific 90th percentiles of a healthy referent subsample to define fatty kidneys. We measured the fat deposition in the renal parenchyma by using MRI dual sequences, and we defined fatty kidneys as constituting cases in which a fate percentage of at least 4% was detected in the parenchyma of the kidneys.

The term “fatty kidney” was first coined in the literature in 1883 [17]. Since then, several studies have tried to assess lipid accumulation in the kidneys [18,19,20]. The true prevalence of fatty kidney disease is unknown because, until now, there was no precise modality for diagnosing fatty kidneys. In addition, various studies have measured fat deposition in different parts of the kidneys [21,22,23,24,25]. Most studies have measured fat deposition in the renal hilum, while some measured it in the pararenal space or the perirenal space, and some measured it in the renal parenchyma. In the present study, we measured fat deposition in the renal parenchyma. We used MRI dual sequences to estimate the fat content in the parenchyma of the kidneys. We defined a cut-off point of 4% fat content in the kidneys to diagnose fatty kidneys. We do not have results of a kidney biopsy that would allow us to show a correlation between our estimation and the actual fat accumulation in the kidneys of our patients. Thus, we cannot be certain that our cut-off point of a 4% fat content in the kidneys is sensitive enough. However, it is clear that we detected patients with abnormal fat accumulation in the kidneys. It is not clear whether we need a 4% fat content in the kidneys to define fatty kidneys or if we can define fatty kidneys with even less fat accumulation. The optimal method for the assessment of renal fat accumulation and the cut-off point for the definition of fatty kidneys remain to be determined.

Fatty kidney disease is associated with hypertension. In an analysis of individuals from the Framingham Heart Study, Foster et al. [16] showed that those with fatty kidneys had a higher odds ratio for hypertension (OR: 2.12; *p* < 0.0001), which persisted after adjusting for body mass index (OR: 1.49; *p* < 0.0001) or visceral adipose tissue (OR: 1.24; *p* = 0.049). In a study by Chughtai HL. et al., patients with a systolic BP > 160 mm Hg or a diastolic BP > 100 mm Hg had higher levels of renal sinus fat, quantified via MRI, than those with a BP less than 160/100 mm Hg [18]. In the present study, we also observed, in those with fatty kidneys, higher levels of systolic BP than in those without fatty kidneys. Fat deposition in the kidneys can increase renal interstitial pressure through the compression of renal veins and lymph vessels in the kidneys [18]. This leads to the activation of the renin angiotensin system and, in the presence of volume expansion, increased sodium reabsorption in the loop of Henle and decreased sodium excretion [20]. Activation of the renin angiotensin system and sodium and volume retention may explain the elevated BP associated with obesity and fatty kidneys [20]. Fatty kidney disease has also been linked to kidney injuries and an increased prevalence of chronic kidney disease. In the present study, we found decreased renal function in those with fatty kidneys. Several mechanisms, such as pro-inflammatory signals, oxidative stress, mitochondrial dysfunction, and dysregulation of serum leptin–adiponectin levels, may be responsible for such kidney injuries [10]. Several clinical trials have demonstrated associations between fatty kidneys and comorbidities including atherosclerosis, cardiovascular events, microalbuminuria, and vascular calcification [18,26,27].

We observed a combination of a fatty liver and fatty kidneys in 6% of the individuals.

The main risk factor for the combination of a fatty liver and fatty kidneys was obesity. We also showed that obese individuals with a fatty liver are at a high risk of having fatty kidneys. The association between obesity and fatty liver disease is well known [28]. However, the association between obesity and fatty kidneys is less known. This association was clearly observed in animal models of obesity. Data from animal studies suggest that obesity leads to kidney damage and hypertension through lipotoxicity, oxidative stress, inflammation, and fibrosis [29,30,31]. Obese mice fed a high-fat diet developed lipid accumulation in the glomeruli and proximal tubules. This lipid accumulation was associated with albuminuria, increased systolic BP and oxidative stress, and a larger glomerular tuft area and mesangial matrix [30]. It is not clear whether obesity itself leads to fatty kidneys or if food ingredients determine whether fat will accumulate in the kidneys. Unfortunately, we do not have data on the dietary content of our cohort. However, we have data from recent animal studies. We showed in a rat model that a high-fructose diet (HFD) can induce metabolic-like syndrome and increase triglyceride accumulation in the kidneys [9]. In this model, oil red staining revealed lipid droplet formation and increased adipophilin mRNA expression in the kidneys, supporting the formation of fatty kidneys. Unlike a HFD, a high-sucrose diet increased BP and caused mild worsening of insulin resistance in spontaneous hypertensive rats, but it did not cause metabolic syndrome [32]. These animal studies suggest that high-fat and high-fructose diets are important components in the development of fatty kidneys.

It is uncertain whether the hypertension and renal impairment associated with obesity are specifically present because of renal fat accumulation or are related to weight gain and adiposity. Through modeling structure and serial adjustment for BMI and visceral adipose tissue, Katharina Mueller-Peltzer et al. showed the independent role of renal fat accumulation in the pathogenesis of hypertension and chronic kidney disease [33].

The question remains whether weight reduction will reduce lipid accumulation in the liver and the kidney. Regarding the liver, two recent studies showed that weight reduction improved parameters of a fatty liver [34,35]. Regarding the kidneys, most dietary weight loss trials failed to show renal benefits. Small studies have shown that dietary weight loss improves albuminuria and attenuates the decline in renal function in patients with chronic kidney disease [36]. Bariatric surgery has yielded data showing that weight loss has a beneficial effect on the kidneys. Weight loss with bariatric surgery normalized hyperfiltration and reduced hypertension and albuminuria [37]. A recent review of bariatric surgery with up to 7 years of follow-up data showed an improvement of chronic kidney disease and less development of end-stage renal disease [38], which supports the concept that ectopic renal fat is worsened by obesity and improved with weight loss, as seen in the case of a fatty liver.

Fatty kidney disease is a common condition in obese individuals and is associated with a fatty liver and an increased risk of hypertension and chronic kidney disease. Fatty kidneys may be associated with BP regulation and chronic kidney disease in humans and provide additional insight into the pathophysiologic role of adiposity in renal dysfunction. Further research is necessary to evaluate the longitudinal associations of fatty kidneys with markers of renal function and metabolic risk factors.

Our study strengthens the association between obesity, a fatty liver, and fatty kidneys.

Our study has certain limitations. The cohort was composed of patients who underwent abdominal MRI for any indication and as such may not accurately represent the general population. While reviewing the scans, we did not take into account their indications. In addition, we had no data on their diets and the drug therapy they received, and therefore we cannot comment on the impact of diet and drug therapy on the development of a fatty liver and fatty kidneys.

Nevertheless, this study provides evidence that reviewing MRI scans of any indication should include measuring fat fractions in the kidneys in addition to the liver. Patients who have been found to have a fatty liver and fat changes in their kidneys should undergo further testing, such as being subjected to a urine protein test and kidney function and BP measurements. This will allow us to track down undiagnosed conditions such as hypertension, diabetes, and dyslipidemia, diseases whose early diagnosis can prevent morbidity.

While we do not yet see the value of ordering an imaging study just to make a diagnosis of fatty kidney disease, we do believe that radiologists reviewing abdominal MRI scans can specifically assess and report ectopic and increased renal fat. A renewed focus on the kidneys will help develop and refine radiologic diagnostic criteria for fatty kidney disease, much like what has been conducted for the liver.

## 5. Conclusions

Obesity is a major risk factor for fatty liver and fatty kidney disease. Individuals with a fatty liver are more likely to have fatty kidneys. MRI is an accurate modality for diagnosing fatty kidney disease. Reviewing MRI scans of any indication should include an assessment of the fat fraction in the kidneys in addition to that of the liver.

## Figures and Tables

**Figure 1 nutrients-16-01387-f001:**
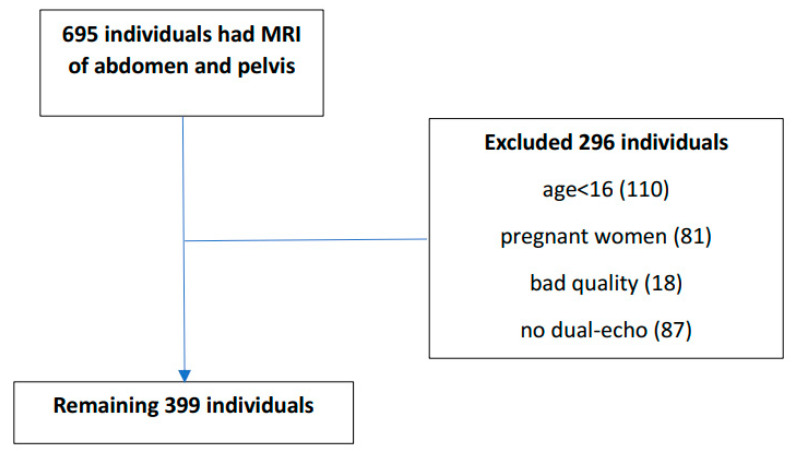
Study population flow chart.

**Figure 2 nutrients-16-01387-f002:**
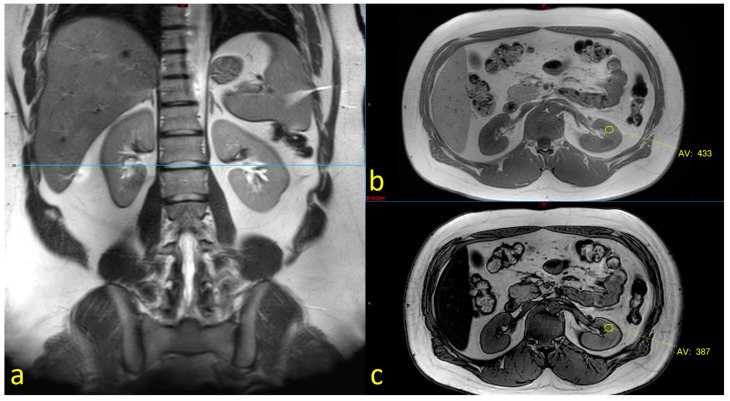
Assessment of renal fat using dual-phase MRI sequences: Measurements were performed on three levels of the kidney as defined on control T2Wis: upper pole, renal hilum (**a**), and lower pole. In-phase (IP) (TR 127.85, TE 2.30) (**b**) and opposed phase (OP) (TR 127.85, TE 1.15) (**c**) images of (**a**) 28 Y male. Assessment of the fat fraction was performed using the following equation: $\eta =\frac{IP-OP}{2IP}$ (Ref. [13]).

**Figure 3 nutrients-16-01387-f003:**
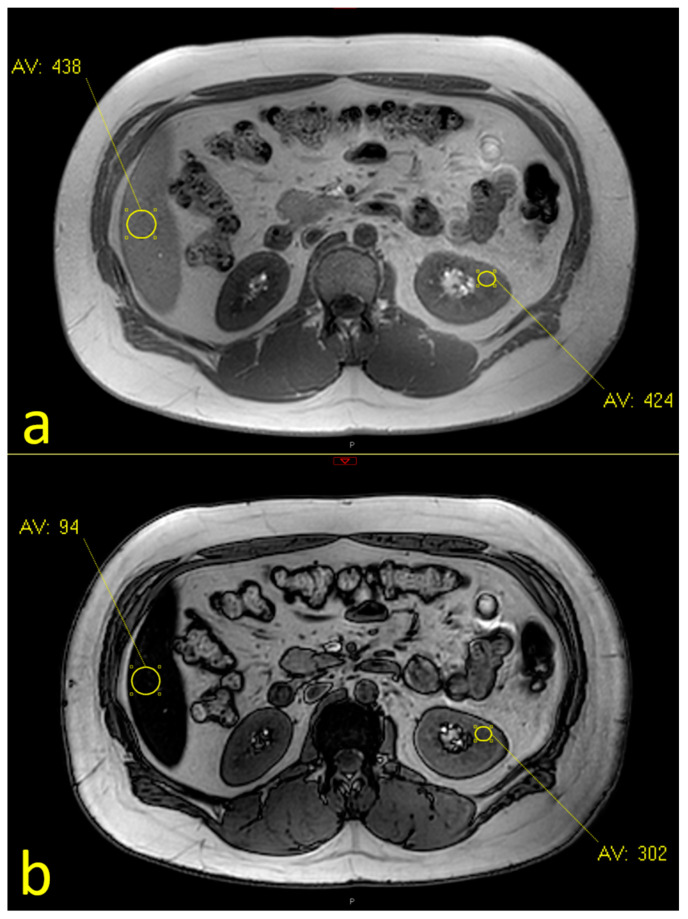
Assessment of renal and liver fat using dual-phase MRI sequences: (Same patient as in Figure 2): In-phase (**a**) (IP) (TR 127.85, TE 2.30) and opposed-phase (OP) (**b**) (TR 127.85, TE 1.15) axial images demonstrating fat fractions of 14% in the lower pole of the kidney and of 39% in the liver.

**Table 1 nutrients-16-01387-t001:** Basic characteristics of the study cohort arranged by fatty liver group.

	Total (*n* = 399)	Fatty Liver (*n* = 72)	Non-Fatty Liver (*n* = 327)	*p*
Female sex, n, (%)	209 (52.4)	34 (47.2)	175 (53.5)	0.33
Age, y, median (IQR)	51 (35–67)	56 (44–67)	49 (33–67)	0.141
Weight, kg, median (IQR) *	72 (60–85)	88 (74–102)	68.5 (58–79)	<0.0001
Height, cm, median (IQR) *	168 (162–175)	167 (162–175)	168 (161–175)	0.800
BMI, kg/$m^{2},$ median (IQR) *	24.6 (21.8–29.3)	30.8 (27.4–34.1)	24.1 (21.3–27.3)	<0.0001
Systolic BP, mmHg, median # (IQR)	121 (110–140)	127 (115–142)	120 (109–138)	0.033
Diastolic BP, mmHg, median # (IQR)	73 (66–80)	77 (72–81)	73 (66–80)	0.017
Hypertension, n, (%)	86 (21.6)	22 (30.6)	64 (19.6)	0.040
Diabetes Mellitus, n, (%)	38 (9.5)	12 (16.7)	26 (8)	0.023
Dyslipidemia, n, (%)	24 (6)	9 (12.5)	15 (4.6)	0.011

Fatty liver—grade 1 and grade 2 steatosis. Non fatty liver—grade 0 steatosis. BMI = Body mass Index; BP = blood pressure. * Missing data for 79 subjects. # Missing data for 52 subjects.

**Table 2 nutrients-16-01387-t002:** Basic characteristics of the study cohort according to fatty kidney group.

	Total (*n* = 382)	Fatty Kidneys (*n* = 71)	Lack of Fatty Kidneys (*n* = 311)	*p*
Female sex, n, (%)	203 (53.1)	31 (43.7)	172 (55.3)	0.076
Age, y, median (IQR)	50 (34–66)	56 (35–70)	49 (34–64)	0.076
Weight, kg, median (IQR) *	71 (60–84)	75 (65–88)	70 (58–83)	0.006
Height, cm, median (IQR) *	168 (160–175)	169 (162–174)	168 (160–175)	0.616
BMI, kg/$m^{2},$ median (IQR) *	24.6 (21.6–29.1)	27.2 (23.1–30.0)	24.4 (21.4–28.7)	0.013
Systolic BP, mmHg, median # (IQR)	120 (110–140)	126 (114–143)	120 (109–137)	0.032
Diastolic BP, mmHg, median # (IQR)	73 (66–80)	76 (66–80)	73 (66–80)	0.464
Hypertension, n, (%)	79 (20.7)	17 (23.9)	62 (19.9)	0.452
Diabetes Mellitus, n, (%)	35 (9.2)	8 (11.3)	27 (8.7)	0.496
Dyslipidemia, n, (%)	20 (5.2)	4 (5.6)	16 (5.1)	0.867

Fatty kidney was defined as ≥4% fatty change. A non-fatty kidney was defined as <4%. BMI = Body mass index; BP = blood pressure. * Missing data for 77 subjects. # Missing data for 52 subjects.

**Table 3 nutrients-16-01387-t003:** Basic characteristics of the study cohort according to fatty kidney and fatty liver groups.

	Fatty Liver (*n* = 67)			Non-Fatty Liver (*n* = 315)		
	Fatty Kidney (*n* = 23, 34.33%)	No Fatty Kidneys (*n* = 44, 65.67%)	*p*	Fatty Kidney (*n* = 48, 15.24%)	Non-Fatty Kidney (*n* = 267, 84.76%)	*p*
Female sex, n, (%)	12 (52.2)	21 (47.7)	0.730	19 (39.6)	151 (56.6)	0.030
Age, y, median (IQR)	56 (45–70)	51 (40.5–61.75)	0.326	54.5 (34–70)	49 (33–65)	0.201
Weight, kg, median (IQR) *	90 (81–101)	85 (72–103)	0.211	72 (61–81)	68 (57–77)	0.182
Height, cm, median (IQR) *	165 (159–178)	168 (162–175)	0.790	170 (165–173)	168 (160–175)	0.375
BMI, kg/$m^{2},$ median (IQR) *	32.0 (29.7–35.6)	29.8 (25.0–23.4)	0.059	24.3 (21.6–27.6)	24.0 (21.0–27.1)	0.394
Systolic BP, mmHg, median (IQR) #	133 (114–156)	120 (115–141)	0.115	125 (111–140)	120 (108–136)	0.104
Diastolic BP, mmHg, median (IQR) #	78 (74–82)	76 (70–80)	0.440	73 (66–78)	72 (65–80)	0.864
Hypertension, n, (%)	8 (34.8)	11 (25)	0.399	9 (18.8)	51 (19.1)	0.955
Diabetes Mellitus, n, (%)	5 (21.7)	5 (11.4)	0.258	3 (6.3)	22 (8.2)	0.639
Dyslipidemia, n, (%)	2 (8.7)	4 (9.1)	0.957	2 (4.2)	12 (4.5)	0.919

BMI = Body mass index; BP = blood pressure. * Missing data for 77 subjects. # Missing data for 52 subjects.

**Table 4 nutrients-16-01387-t004:** Adjusted odds ratio for fatty kidney.

Parameter	Odds Ratio	CI 2.5%	CI 97.5%	*p*-Value
Age	1.0	1.0	1.0	0.086
Sex	1.6	0.9	2.7	0.082
Diabetes Mellitus	1.0	0.4	2.5	0.987
Hypertension	0.9	0.4	1.8	0.688
Dyslipidemia	0.8	0.2	2.7	0.722
Fatty Liver	2.9	1.6	5.2	0.001

**Table 5 nutrients-16-01387-t005:** The rates of fatty liver and fatty kidney disease according to body mass index.

	Total (*n* = 322)	BMI ≤ 25 (*n* = 175, 54.3%)	25 < BMI < 30 (*n* = 84, 26.1%)	BMI ≥ 30 (*n* = 63, 19.6%)	*p* Value
Female sex, n, (%)	166 (51.6)	91 (52)	40 (47.6)	35 (55.6)	0.625
Age >50 y, n (%)	166 (51.6)	74 (42.3)	49 (58.3)	43 (68.3)	0.0006
Systolic BP > 130 mmHg, n (%)	116 (36.0)	44 (25.1)	33 (39.2)	39 (62.0)	<0.0001
Diastolic BP > 80 mmHg, n (%)	74 (23.0)	37 (21.1)	18 (21.4)	19 (30.2)	0.319
Hypertension, n, (%)	82 (25.5)	27 (15.4)	23 (27.4)	32 (50.8)	<0.0001
Diabetes Mellitus, n, (%)	37 (11.5)	13 (7.4)	7 (8.3)	17 (27)	<0.0001
Dyslipidemia, n, (%)	23 (7.1)	8 (4.6)	7 (8.3)	8 (12.7)	0.088
Fatty liver, n, (%)	57 (17.7)	7 (4)	18 (21.4)	32 (50.8)	<0.0001
Fatty kidney, n, (%)	56 (17.4)	22 (12.6)	22 (26.2)	12 (19)	0.007
Fatty liver and fatty kidney, n, (%)	17 (5.3)	0 (0)	7 (8.3)	10 (15.9)	<0.0001

BMI = body mass index; BP = blood pressure. Differences in baseline characteristics between the groups were tested using the chi-squared test.

**Table 6 nutrients-16-01387-t006:** A: Adjusted OR for fatty kidneys. B: Adjusted OR for fatty kidneys + fatty liver.

**A**				
**Parameter**	**Odds Ratio**	**CI 2.5%**	**CI 97.5%**	***p*-Value**
Age	1.0	1.0	1.0	0.086
Sex	1.6	0.9	2.7	0.082
Diabetes mellitus	1.0	0.4	2.5	0.987
Hypertension	0.9	0.4	1.8	0.688
Dyslipidemia	0.8	0.2	2.7	0.722
Fatty Liver	2.9	1.6	5.2	0.001
**B**				
**Parameter**	**Odds Ratio**	**CI 2.5%**	**CI 97.5%**	***p*-Value**
Age	1.0	1.0	1.0	0.958
Sex	0.6	0.2	1.8	0.359
Diabetes mellitus	1.7	0.4	6.8	0.478
Hypertension	1.3	0.3	4.8	0.694
Dyslipidemia	0.5	0.1	4.9	0.545
BMI ≥ 30 kg/m^2^	6.3	2.1	18.6	0.001

BMI = Body mass index.

## Data Availability

Data are not available.

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
