# Peer review of "Obesity Is Associated with Fatty Liver and Fat Changes in the Kidneys in Humans as Assessed by MRI"

_nutrients, 2024, doi:10.3390/nu16091387_

Round 1

Reviewer 1 Report

Comments and Suggestions for Authors

Authors presented manuscript on the incidence of fatty liver and fatty kidney in patients undergoing MRI scans. While the idea is interesteng I have some thoughts regarding the execution of the study.

Editorial correction is necessary due to a number of punctuation errors (e.g.: melli-tus, or-gans)

Unfortunately there are no data on renal function (at least to some extent there might be creatinine level and therefore eGFR). In many cases prior to contrast-enhanced MRI it is necessary. Are there any markers for kidney function damage available?

Patients with BMI <18,5 should be excluded from the analysis

Lack of criteria of “dyslipidemia”

Statistics:

Subgroups according to BMI should have slightly different cut-off points:

  • BMI 25.0 to <30 means overweight.
  • BMI 30.0 or higher means obesity 

There is no data on the assessment of normality of distribution of variables. The t-test was used (that would suggest normal distribution), but data are presented with median and IQR - data should be presented as mean +/- 95%CI or SD.

Lack of description of statistical tool used to assess groups in tab. 5 (multiple comparisons). Io the bottom of table it states ANOVA, its not mentioned in statistics section and no in between groups differences are shown (no post-hoc tests are reported).

Therefore complete data on statistics should be added as supplementary file including degrees of freedom t-test actual values

In summary

Data are not clearly presented in the view of their completeness – there is information about missing BMI data (around 20 % of study population). It is important because  for example BMI of total group is reported as 24.6 in 399 subjects as well as 322 subjects, which is confusing. Other missing data should also be mentioned. In my opinion in this case all patients without BMI – which is crucial for the concept of the study should be excluded from analysis. Furthermore there are many confounders that have not been addressed  current treatment and drugs used (antihypertensive, antidiabetic, hipolipidemic). Also, indications for MRI scans are not mentioned, which might also affect data interpretation

Comments on the Quality of English Language

Minor spelling mistakes

Author Response

Reviewer 1

Editorial correction is necessary due to a number of punctuation errors (e.g.: melli-tus, or-gans).

We corrected the punctuation errors in the revised manuscript

Unfortunately there are no data on renal function (at least to some extent there might be creatinine level and therefore eGFR). In many cases prior to contrast-enhanced MRI it is necessary. Are there any markers for kidney function damage available?

We thank the reviewer for his comment we have the creatinine and the urea of all participants and indeed the creatinine was higher in those with fatty kidney than in those without fatty kidney. (0.89 vs. 0.76 and 35 vs 30 for the creatinine and urea respectively p < 0.05 for both). This was included in the results section of the revised manuscript.  

Patients with BMI <18,5 should be excluded from the analysis.

We identified 18 patients with BMI < 18.5 and we analyzed the data with this group and without this group and the message was the same, therefore we prefer to leave the groups as they are.

 Lack of criteria of “dyslipidemia”

The criteria of Dyslipidemia was either use of lipid lowering agents or ICD-9 code of the diagnosis in the chart. This was included in the revised manuscript 

Statistics:

Subgroups according to BMI should have slightly different cut-off points:

  • BMI 25.0 to <30 means overweight.
  • BMI 30.0 or higher means obesity

We thank the reviewer for his comment. We have double-checked the results and we did not have subjects with BMI of exactly 30.00, so we changed the definition in the text and the table, but the numbers remained the same.

There is no data on the assessment of normality of distribution of variables. The t-test was used (that would suggest normal distribution), but data are presented with median and IQR - data should be presented as mean +/- 95%CI or SD.

We thank the reviewer for his comment. We used, in the revised manuscript, the non-parametric Mann-Whitney U test for paired comparisons. The p values were corrected in the new tables. This was also corrected in the statistical analysis section of the revised manuscript.

Lack of description of statistical tool used to assess groups in tab. 5 (multiple comparisons). Io the bottom of table it states ANOVA, its not mentioned in statistics section and no in between groups differences are shown (no post-hoc tests are reported).

Therefore complete data on statistics should be added as supplementary file including degrees of freedom t-test actual values

We thank the reviewer for his comments. For clarity, we deleted the height, weight and BMI values from table 5, since the groups were different per definition. We changed the presentation in the table to categorical variables and we used the chi-square test to compare between the groups. Table 5 has been modified accordingly in the revised manuscript.  

In summary

Data are not clearly presented in the view of their completeness – there is information about missing BMI data (around 20 % of study population). It is important because  for example BMI of total group is reported as 24.6 in 399 subjects as well as 322 subjects, which is confusing. Other missing data should also be mentioned. In my opinion in this case all patients without BMI – which is crucial for the concept of the study should be excluded from analysis. Furthermore there are many confounders that have not been addressed  current treatment and drugs used (antihypertensive, antidiabetic, hipolipidemic). Also, indications for MRI scans are not mentioned, which might also affect data interpretation

The study has two goals. One is to use the new MRI method to diagnose fatty kidney and fatty liver and to calculate the rate of these findings and to evaluate the association between the distinct conditions. For this purpose, we included all 399 patients. The second goal was to evaluate the association between obesity and fatty liver and fatty kidney. For this purpose, we could analyze only 322 patients. Therefore, we left the tables as they are and added notes on the missing data for BMI and blood pressure levels in the first 3 tables.

Unfortunately, we do not have sufficient data on the treatment and drug used and this is included in the limitation of the study. In most cases, the indications for MRI scans were evaluation of suspected malignancy or follow-up after removal of abdominal or gynecological tumor. This statement was included it in the methods section of the revised manuscript.

Comments on the Quality of English Language

Minor spelling mistakes

We corrected the spelling mistakes.

Reviewer 2 Report

Comments and Suggestions for Authors

Dear Authors,

Thank you for the opportunity to review this paper. The study is very well conducted and the statistical tests are robust.

Please find my comments below:

Introduction

Line 66 Please write more clearly the aim of the study such as:  The aim of the study was to use abdominal  MRI scans/technique…etc. What novelty does  this study have? Please underline.

Methodology

Line 75 The weight and height are self reported?

Results

Line 124 Please add median age± SD

Line 125 median BMI ± SD, please add the standard deviation (SD)

Discussion

The authors discussed the possibility that weight reduction will reduce the fatty kidney. I think that they must also discuss if weight reduction may reduce the fatty liver.

Conclusion. Please write the conclusion of the present study.

Kind regards

Author Response

Reviewer 2

Introduction

Line 66 Please write more clearly the aim of the study such as:  The aim of the study was to use abdominal  MRI scans/technique…etc. What novelty does  this study have? Please underline.

The novelty of the study was the use of a new MRI-based assessment of kidney fat fraction using dual-phase sequences Methodology

We wrote more clearly the aim of the study and emphasized the use of the new method to assess fat accumulation in the kidney.

Line 75 The weight and height are self reported

The weight and height were retrieved from the charts. We believed that in most cases it was self reported.

Results

Line 124 Please add median age± SD

We added the mean ± SD

Line 125 median BMI ± SD, please add the standard deviation (SD)

We added the mean ± SD

Discussion

The authors discussed the possibility that weight reduction will reduce the fatty kidney. I think that they must also discuss if weight reduction may reduce the fatty liver.

We added in the discussion a statement on the effect of weight reduction on fatty liver.  

Conclusion. Please write the conclusion of the present study.

We added a conclusion of the study